# Splicing Dysregulation as Oncogenic Driver and Passenger Factor in Brain Tumors

**DOI:** 10.3390/cells9010010

**Published:** 2019-12-18

**Authors:** Pamela Bielli, Vittoria Pagliarini, Marco Pieraccioli, Cinzia Caggiano, Claudio Sette

**Affiliations:** 1Department of Biomedicine and Prevention, University of Rome Tor Vergata, 00133 Rome, Italy; 2Laboratory of Neuroembryology, IRCCS Fondazione Santa Lucia, 00143 Rome, Italy; mar.pieraccioli@gmail.com (M.P.); cinzia.caggiano@virgilio.it (C.C.); 3Department of Neuroscience, Section of Human Anatomy, Catholic University of the Sacred Heart, 00168 Rome, Italy; vittoria.pagliarini@unicatt.it; 4IRCCS Fondazione Policlinico Universitario A. Gemelli, 00168 Rome, Italy

**Keywords:** alternative splicing, brain tumors, splicing factors, EGFR signaling, hippo signaling, tumor microenvironment, PRMT5

## Abstract

Brain tumors are a heterogeneous group of neoplasms ranging from almost benign to highly aggressive phenotypes. The malignancy of these tumors mostly relies on gene expression reprogramming, which is frequently accompanied by the aberrant regulation of RNA processing mechanisms. In brain tumors, defects in alternative splicing result either from the dysregulation of expression and activity of splicing factors, or from mutations in the genes encoding splicing machinery components. Aberrant splicing regulation can generate dysfunctional proteins that lead to modification of fundamental physiological cellular processes, thus contributing to the development or progression of brain tumors. Herein, we summarize the current knowledge on splicing abnormalities in brain tumors and how these alterations contribute to the disease by sustaining proliferative signaling, escaping growth suppressors, or establishing a tumor microenvironment that fosters angiogenesis and intercellular communications. Lastly, we review recent efforts aimed at developing novel splicing-targeted cancer therapies, which employ oligonucleotide-based approaches or chemical modulators of alternative splicing that elicit an impact on brain tumor biology.

## 1. Introduction

Tumors of the central nervous system (CNS) are a heterogeneous group of diseases that are classified on the basis of molecular parameters and histopathological characteristics. A comprehensive classification of CNS tumors was first established by the World Health Organization (WHO) in 2016 [1], substituting previous ones that were based only on histological features. The incorporation of well-established genetic parameters in the new guidelines has particularly improved the classification of diffuse gliomas and embryonal tumors, such as medulloblastoma [1]. While the identification and validation of further objective molecular parameters are still necessary to improve the accuracy of the current classification [1], changes made so far have highlighted the importance of elucidating molecular aspects of CNS tumors to better stratify patients. Among CNS tumors, herein we mainly focus on gliomas and medulloblastomas, for which a clear involvement of splicing dysregulation has been described. 

Diffuse gliomas represent a broad category of brain and spinal cord tumors that arise from glial cells. According to the WHO classification, they are subdivided in grade II and III astrocytomas, grade II and III oligodendrogliomas and grade IV glioblastomas (Table 1) [1]. 

Glioblastoma (GBM) arises from the astrocytic lineage [1] and accounts for more than 60% of all brain tumors in adults [2]. It is associated with very poor prognosis and displays a median survival of one year [3]. Current treatments include tumor resection, radiation and chemotherapeutic treatment with temozolomide [4]. Despite these invasive therapies, GBMs almost invariably relapse due to their highly resistant and infiltrative nature [5]. Several prognostic biomarkers allow stratification of GBM patients. For instance, the methylation status of O6-methylguanine methyltransferase (*MGMT*) gene promoter predicts benefit from temozolomide chemotherapy. Other molecular markers used in patient stratification are the mutational status of the isocitrate dehydrogenase (*IDH*) and epidermal growth factor receptor (*EGFR*) genes [6]. *IDH* mutations are found in ~5–10% of GBM patients and are associated with better outcomes [7]. On the other hand, ~25% of GBM patients that are wild-type for *IDH* show *EGFR* amplification, with half of these tumors harboring a deletion in the gene that yields a receptor endowed with ligand-independent activity [8]. 

Grade IV medulloblastoma (MB, also referred to as primitive neuroectodermal tumor) arises from the primitive or embryonal cells of the cerebellum (Table 1). MB is the most common malignant brain tumor in children [9] with a culminating incidence before the age of five [10]. They are treated using combined therapeutic approaches, which include surgical resection as well as radio- and chemo- therapy [11]. Although these treatments strongly improve overall survival rates of MB patients, they are extremely aggressive and devastating for these young patients. MBs are subdivided into four subgroups on the basis on distinct molecular parameters, namely wingless-activated (WNT), sonic hedgehog-activated (SHH), Group 3, and Group 4 MBs [12]. Among them, WNT MB displays the most favorable prognosis, whereas Group 3 shows the worst prognosis, and SHH and Group 4 MBs are characterized by an intermediated clinical course [13,14]. 

## 2. Dysregulation of Splicing in Human Cancers 

Besides genetic alterations, the deregulation of gene expression programs is a well-established hallmark of cancer [15]. The aberrant expression of oncogenic and tumor-suppressor transcription factors often underlies the alteration of transcriptional and post-transcriptional mechanisms in cancer cells. Furthermore, the dysregulation of splicing regulatory mechanisms has clearly emerged as a typical feature of cancer cells, which can either drive or contribute to tumor onset and progression [16,17]. 

High-throughput sequencing studies in the last decade have shown that nearly all human genes undergo alternative splicing regulation [18,19]. This mechanism is clearly advantageous for eukaryotic cells as it endows them with a plastic genome and contributes to the expansion of their proteome diversity. Indeed, splicing choices can be fine-tuned in the cell by the activation of signal transduction pathways in response to both external and endogenous cues. In this way, cells modulate their proteome to withstand changes in the surrounding environment, or in response to the activation of a specific differentiation program [20]. Nevertheless, its remarkable flexibility represents a risk factor, and splicing dysregulation concurs with many human diseases [20] including cancer [16,17]. The aberrant splicing events occurring in cancer cells often lead to generation of protein variants displaying altered function, which can contribute to tumorigenesis [16]. Failure to properly recognize a splice site is often the result of mutations in the splice site sequence and/or of the dysregulation of splicing factor expression in cancer cells. Moreover, mutations in genes encoding core proteins and RNAs of the splicing machinery have recently emerged as oncogenic drivers which promote widespread transcriptome modifications [16,21]. Herein, we review the current knowledge concerning splicing dysregulation in brain tumors, with particular attention paid to pro-oncogenic processes underlying cancer onset and progression. 

## 3. Regulation of Alternative Splicing 

Splicing is operated by a ribonucleoprotein complex, named spliceosome, which catalyzes two successive trans-esterification reactions at the exon–intron boundaries, thus removing introns and joining adjacent exons [22]. Canonical consensus sequences, including the 5′ and 3′ splice sites (ss), the branch point and the polypyrimidine tract, contribute to exon recognition by driving stepwise recruitment of spliceosomal small nuclear ribonucleoproteins (U1, U2, U4, U5, and U6 snRNPs). The base pairing of the U1 small nuclear RNA (snRNA) to the 5′ss and the binding of splicing factor 1 (SF1) to the branch point are the first events. The successive recruitment of the U2AF65/U2AF35 dimer to the polypyrimidine tract and to the 3′ss, respectively, facilitates the replacement of SF1 with the U2 snRNP. This complex in turn recruits U4/U6-U5 snRNPs and promotes the formation of catalytically active spliceosome complexes that mediate trans-esterification reactions [23]. However, since the sequences recognized by the spliceosome are short and degenerate, and human exons are generally much shorter (~150 bp) than introns (>1000 bp), proper exon recognition is a difficult task that requires additional *cis*-regulatory RNA elements and *trans*-acting factors [22,24]. Moreover, competition between *trans*-acting factors for binding to *cis*-regulatory RNA elements can result in the alternative selection of exons in the mature mRNA. Indeed, while many exons are constitutively spliced, nearly all human genes also contain variant exons that can be alternatively spliced [18,19], yielding mRNA species that could present a distinct regulatory fate [25] and/or give rise to multiple protein products from a single gene [26,27]. Alternative splicing events can affect the entire exon (cassette exon) or part of it, through the usage of alternative 5′ and 3′ ss that are present in a variant exon. Furthermore, mutually exclusive exons can be alternatively spliced, while introns can be retained in the mature transcript. Lastly, alternative first exons can be selected in genes bearing multiple transcription start sites, whereas the usage of multiple polyadenylation sites can yield alternative last exons [17]. The combination of one or more of these alternative splicing types allows each human gene to encode multiple splice variants and proteins, thus enhancing the coding potential of the genome [20,22,24]. 

The *cis*-regulatory RNA elements can be distinguished in exonic or intronic splicing enhancers or silencers. These sequences are recognized by *trans*-acting splicing factors that modulate the recruitment of spliceosome components near the target exon. The major families of proteins involved in splicing regulation are the serine-arginine rich (SR) proteins [28] and the heterogeneous nuclear ribonucleoproteins (hnRNPs) [29]. SR proteins preferentially bind enhancer elements and exhibit a positive role in splicing regulation. Conversely, the binding of hnRNPs to silencer elements generally represses exon inclusion. SR proteins and hnRNPs can compete for binding to *cis*-regulatory RNA elements and/or for the recruitment of spliceosome component(s). Furthermore, the activity of many splicing factors is regulated by the reversible phosphorylation mediated by specific protein kinases, which fine-tunes their activity in response to the activation of signaling pathways [30]. Phosphorylation impacts most features of splicing factors, from the affinity for their target RNA, to the regulation of their subcellular localization and interaction with co-factors [30]. Moreover, alternative splicing is also influenced by the context of the *cis*-regulatory RNA elements [31,32,33,34,35], RNA structure [36], transcription rate, and the chromatin epigenetic signature [37,38]. 

## 4. Splicing Dysregulation in Brain Tumors

Deregulation of the activity and/or expression of several splicing factors in brain tumors was shown to elicit a significant impact on their transcriptome [39,40,41,42]. Current knowledge mainly relates to GBM, the most common type of adult brain tumors. For instance, the spliceosome component SmB/B’ (SNRPB) is up-regulated and exerts a pro-oncogenic role in GBM by controlling the expression and splicing of several genes associated with gliomagenesis [42] (Table 2). 

Similarly, the high expression of PTBP1 (also known as hnRNPI) and/or its amplification in GBM maintain progenitor-specific splicing patterns that counteract tumor suppressor functions and favor tumor progression [43]. Among other events, PTBP1 promotes the aberrant splicing of Brahma/SWI2-related gene 1 (BRG)-associated factor (BAF45d) [44], a co-transcriptional regulator involved in brain development (Table 2). Importantly, the proto-oncogene c-MYC was shown to orchestrate the concomitant up-regulation of PTBP1 and two other hnRNPs (hnRNPA1, hnRNPA2) in GBM [45], which cooperate in the splicing regulation of the pyruvate kinase gene (*PKM*) [45,46] (Table 2). PKM catalyzes the last step of the glycolytic pathway by promoting the conversion of phosphoenolpyruvate to pyruvate. Mutually exclusive splicing of exon 9 or 10 in *PKM* yields either the PKM1 or the PKM2 isoform, respectively. PKM1 is the adult isoform and promotes oxidative phosphorylation, whereas PKM2 is prevalently expressed during embryogenesis and promotes aerobic glycolysis. The binding of hnRNPA1, hnRNPA2, and PTBP1 to sequences flanking exon 9 promotes the inclusion of exon 10 and expression of PKM2, thus ensuring high glycolytic flux in GBM cells [45,46]. Another splicing factor contributing to this metabolic switch is the SR protein SRSF3, which is also up-regulated in gliomas and sustains glioma stem cells (GSCs) growth, self-renewal, and tumorigenicity [47]. Whole transcriptome analysis revealed a role for SRSF3 as widespread modulator of gene expression and splicing in GBM (Table 2). SRSF3 also promotes exon 10 inclusion and PKM2 expression [48]. Notably, reverting this splicing switch by transducing an exon 10-targeting antisense oligonucleotide (ASO) induced apoptosis in GBM cells [49] (Table 2), suggesting their addiction to aerobic glycolysis. 

Splicing regulation can also result in opposite functions of the cancer-specific protein variant. For instance, the cyclin-dependent kinase (Cdk)-associated protein phosphatase KAP dephosphorylates CDK2 and inhibits cell cycle progression. However, its aberrant splicing in GBM cells yields a dominant negative KAP variant that increases proliferation and migration, thus contributing to malignancy [50] (Table 2). Likewise, the up-regulation of hnRNPH in GBM alters the splicing of the anti-apoptotic isoform MAP-kinase activating death domain protein (MADD), thus switching the function of the TNF-α/TRAIL-induced pathway from apoptosis to proliferation [51] (Table 2). On the other hand, SRSF1 promotes splicing of the membrane-localized full-length isoform of class I Myosin 1B (MYO1B-FL) over a cytoplasmic truncated isoform (MYO1B-T) [35] (Table 2). Interestingly, while MYO1B-FL promotes cell proliferation, survival, and invasion, MYO1B-T lacks these pro-oncogenic features. This antagonistic behavior of MYO1B splice variants has been ascribed to the ability of MYO1B-FL to directly recruit the phosphoinositide 3 kinase (PI3K) at the plasma membrane, thus activating the PDK1/AKT and PAK/LIMK signaling pathways [35]. Since up-regulation of SRSF1 associates with tumor progression and poor prognosis in gliomas [35], these observations suggest that the SRSF1-guided splicing switch contributes to malignancy by fueling proliferative pathways. 

In addition to mutations in protein coding genes, recent evidence suggests that the dysregulation of splicing in human cancers may also arise from mutations in non-coding RNAs of the spliceosome. Indeed, mutations in U1 snRNA have been identified across several cancer types and were suggested to act as oncogenic drivers [21]. For instance, 28 of the 164 nucleotides (nt) present in the U1 snRNA are recurrently mutated in bladder cancer. On the other hand, the third nt in U1 snRNA is mutated in 97% and 25% of cases of adult and adolescent SHH-MB, respectively, and the mutation correlates with poor prognosis [52]. Recognition of 5′ss by the U1 snRNP is the first event occurring during splicing [23]. It requires base pairing between a stretch of 8 nt (3-10 nt) of the U1 snRNA with a complementary region at the 5′ss (from +6 to -2 nt from 5′ss) of the pre-mRNA. In this context, the A in the third position (nt 3) of U1 snRNA anneals with the U at +6 position in the intron of the pre-mRNA (Figure 1). 

Thus, the A>G transition at the third nt of the mutated U1 snRNA favors G-C base-pairing and increases the occurrence of cryptic alternative 5′ splicing events and/or intron retention. Interestingly, such a mutation correlated with the splicing-dependent inactivation of the *PTCH1* tumor-suppressor gene and activation of the *GLI2* and *CCND2* oncogenes in SHH-MB, supporting the biological relevance of U1 snRNA mutation in this tumor [52].

Although splicing dysregulation in cancer is generally believed to promote tumorigenesis and more aggressive phenotypes, it may also represent a targetable vulnerability. In this regard, recent findings have highlighted that GBM cells accumulate transcripts with retained introns, which require high activity of the arginine methyl-transferase PRMT5 to be properly spliced. Inhibition of PRMT5 negatively impacted GBM growth by impairing intron removal in genes involved in proliferation [53]. Thus, dependency on PRMT5-induced splicing may represent a vulnerability of GBM that can be exploited therapeutically (see below). This concept may also hold true for other MYCN-amplified brain tumors, as amplification of MYC generally correlates with increase intron retention and higher dependency on the efficiency of the splicing machinery [54].

## 5. Biological Relevance of Alternative Splicing Regulation in Brain Tumors

Cancer-related splicing dysregulation often alters the activity of genes governing a broad spectrum of physiological processes, thus sustaining cell behaviors that predispose to cancer onset and progression. Cellular oncogenic features can evolve from a combination of abnormalities, which affect key signaling pathways and interaction with the tumor microenvironment. Herein, we summarize some examples of how aberrant splicing can impact these processes in brain tumors.

### 5.1. EGFR Signaling Pathway 

The EGFR, also known as ErbB1/Her1 [55], is a key factor for brain tumors as it supports the proliferation and stemness of neural progenitor cells from which these tumors originate. Upon EGF binding, the EGFR undergoes conformational changes and autophosphorylation, thus favoring the binding of the regulatory subunit of PI3K and activation of a signaling cascade that leads to phosphorylation and activation of AKT [56]. In turn, the PI3K/AKT pathway plays a pivotal role in the regulation of key biological processes in brain tumors, including cell proliferation, metabolism, survival, migration and angiogenesis [57]. The *EGFR* gene is often amplified in GBM patients, with a subgroup of them showing aberrant expression of an oncogenic variant named EGFRvIII [58]. This variant arises from an in-frame genomic deletion of the exons 2–7 encoding for the extracellular ligand binding domain. As a consequence, EGFRvIII displays constitutive activation, high oncogenic potential and correlates with poor prognosis [58,59,60,61]. Transcriptomic analysis performed in GBM cells reveals that aberrant EGFR/EGFRvIII signaling leads to up-regulation of the expression of genes that are target of the c-MYC oncogene through a complex and interconnected network [62]. For instance, up-regulation of hnRNPA1 promotes splicing of a truncated isoform of MAX (ΔMAX), a core MYC cofactor. ΔMAX enhances the expression of glycolytic genes and favors glucose uptake, glucose-dependent cell proliferation and GBM growth [62] (Figure 2 and Table 2). The expression of hnRNPA1 positively associates with that of EGFRvIII and c-MYC-regulated glycolytic genes in GBM patients and this signature correlates with short overall survival [62]. These findings suggest that aberrant EGFR signaling promotes splicing dysregulation, which in turn amplifies EGFR signaling and metabolic changes that support GBM growth.

Inhibition of EGFR degradation also contributes to malignancy in brain tumors [63]. Upon endocytic internalization, EGFR is trafficked from early endosomes to late endosomes/lysosomes for degradation by the cargo transport GTPase protein Rab7A. Alternatively, EGFR is recycled to the cell membrane to maintain EGFR signaling [63]. Alteration of the balance between degradation and recycling affects EGFR signaling in physiological and pathological conditions. Notably, the up-regulation of PTBP1 in GBM alters the splicing of Annexin A7 (ANXA7), a membrane-bound tumor suppressor protein involved in endosomal organization and function [64]. PTBP1 promotes the skipping of exon 6 in ANXA7 and expression of this isoform diminishes the endosomal internalization of EGFR (Figure 2), thus enhancing signaling during tumor progression [43] (Table 2).

Alternative splicing of the *CD44* gene also modulates EGFR degradation in GBM. CD44 is a transmembrane glycoprotein receptor that controls cell proliferation, adhesion, migration, signaling, and survival. The *CD44* gene comprises 20 exons, with 10 of them being regulated by alternative splicing. The standard CD44 isoform (CD44s) contains only the constitutive exons, whereas multiple CD44 variant isoforms (CD44v) derive from differential assortments of the other alternatively spliced exons [65]. In GBM, CD44s, but not CD44v, attenuates EGFR degradation, resulting in enhanced AKT signaling. CD44s internalizes from cell surface to the endosomal compartments and inhibits Rab7A-mediated EGFR degradation [66] (Figure 2 and Table 2). Accordingly, CD44s expression correlates with highly invasive behavior and poor prognosis [67].

### 5.2. Hippo Signaling Pathway

The Hippo pathway is a key regulator of organ size and tissue homeostasis during development [68]. Activation of the Hippo pathway promotes phosphorylation of the large tumor suppressor homolog 1/2 (LATS1/2) by the mammalian Ste20-like kinases 1/2 (MST1/2). Once activated, LATS1/2 phosphorylates the yes-associated protein (YAP) and the transcriptional co-activator with PDZ binding motif (TAZ), which are the key effectors of the pathway. Phosphorylation of the YAP/TAZ complex sequesters it in the cytoplasm and targets it for degradation, thus halting cell proliferation. Conversely, Hippo pathway inhibition leads to nuclear translocation of YAP/TAZ and activation of a transcriptional program guided by the TEAD family transcription factors that sustains cell proliferation [68]. Deregulation of the Hippo pathway contributes to development of several cancers [69], including brain tumors [70]. For instance, *YAP* gene amplification was shown to contribute to SHH-MB [71], whereas *YAP* gene fusions with *MAMLD1* or *C11orf95* have been identified in ependymal tumors. Since neural stem cells carrying the *YAP-C11orf95* fusion gene form brain tumors when grafted into mice, such genetic aberrations were proposed to promote tumorigenesis [72]. 

Splicing factors can impact the expression of components of the Hippo pathway in brain tumors. For instance, the ubiquitin specific protease 39 (USP39), promotes RNA processing and expression of TAZ, thus favoring glioma oncogenic features in vitro and in vivo [73] (Table 2). Furthermore, recent evidence indicates that gliomas express high levels of a splice variant of the prostate transmembrane (TM) protein, androgen induced 1 (PMEPA1) called PMEPA1a, which promotes degradation of the tumor suppressor LATS1 protein. The decreased levels of LATS1 unleash YAP activity and support pro-oncogenic features in glioma cells [74] (Table 2). In line with these observations, the splicing factor USP39 is up-regulated in high-grade gliomas [73], whereas reduced levels of LATS1 associate with poor prognosis in glioma patients [75]. Although the mechanisms driving these splicing-related events (i.e., up-regulation of USP39 and splicing of PMEPA1a) are still unknown, these studies suggest that splicing dysregulation impacts a key developmental program by altering the equilibrium of the Hippo pathway during gliomagenesis. 

### 5.3. Tumor Microenvironment 

Crosstalk between tumor cells and the associated tumor microenvironment (TME) strongly influences cancer initiation, progression, and prognosis [76]. TME comprises several infiltrating fibroblasts and immune cells as well as extracellular components (cytokines, growth factors, hormones, extracellular matrix, etc.) that influence tumor cell biology. Notably, brain tumors hardly colonize outside the CNS, suggesting that the TME establishes a tumor niche that sustains tumor growth and progression. Another peculiarity of brain TME is the presence of a blood-brain barrier, which affects the composition of the extracellular matrix (ECM) as well as the population of resident cells [77]. Mounting evidence indicates that regulation of alternative splicing contributes to the establishment of a specific TME in brain tumors and affects the outcome of the disease. 

In the brain, the main constituents of ECM include proteoglycans, glycoproteins, and glycosaminoglycans, as heparan sulfate proteoglycans (HSPGs) and hyaluronic acid (HA) [77]. HSPGs sequester angiogenic heparin-binding growth factors, such as fibroblast growth factors (FGFs) and vascular endothelial growth factors (VEGFs), which are subsequently released by heparanase (HSPE), thus modulating their availability in the TME [78]. Up-regulation of HPSE in GBM has been associated to the expression of a truncated splice variant of the transcription factor GLI1 [79,80], the terminal effectors of the SHH pathway [81]. Alternative splicing of human GLI1 pre-mRNA yields either the full length isoform or two shorter isoforms named GLI1DN [82] and tGLI1 [80]. The tGLI1 splice variant contains a small in-frame deletion and drives a different transcriptional program with respect to full-length GLI1 [80]. Indeed, tGLI1 sustains GBM angiogenesis and growth by promoting the expression of VEGFA and VEGF receptor 2 (VEGFR2) in both GBM and MB [79] (Table 2). Furthermore, tGLI1 is predominantly activated in the highly aggressive mesenchymal subtype of GBM and in GSCs, suggesting its contribution to the acquisition of a more malignant GBM phenotype [83]. 

The high vascularity that characterizes brain tumors is affected by the splicing dysregulation of VEGF variants. The VEGFA gene consists of eight exons and its alternative splicing yields numerous isoforms displaying different ability to bind heparin and heparin sulphate proteoglycans located on the cell surface and the ECM [84]. In particular, the usage of alternative 3’ss in VEGFA exon 8 generates either the pro-angiogenic VEGF_XXX_a (usage of distal 3’ss) or the anti-angiogenic VEGF_XXX_b (usage of proximal 3’ss) variant (where xxx represents the amino acid number of mature protein) [84]. Moreover, the presence of an alternative translation initiation CUG codon leads to the synthesis of long VEGF (L-VEGF) [84]. Two splice variants of L-VEGF (L-VEGF_144_a and L-VEGF_138_a) have been described in GBM [85,86]. The L-VEGF_144_a variant lacks part of exon 1 and exons 2 to 5, retains exons 7 and 8, and utilizes an in frame alternative intronic 3’ss of exon 6a. This variant was found in GBM patients and it is associated with worse response rate to combined treatments including bevacizumab, temozolomide and radiotherapy (Table 2). Notably, L-VEGF_144_a expression was not associated with different outcome in patients receiving the standard therapy (temozolomide and radiotherapy) [86], suggesting that the VEGF epitope targeted by bevacizumab is absent in L-VEGF_144_a. Thus, the different response to these therapies indicates that expression of this splice variant could be used as prognostic factor for bevacizumab treatment, a currently Food and Drug administration (FDA)-approved therapy for GBM [87]. 

Splicing regulation also affects the reciprocal interaction between tumor cells and TME during migration and invasion [88]. The long splice variant of the scaffold protein Intersectin 1 (ITSN1-L) is highly enriched in neurons, whereas the short ITSN1-S isoform is mainly expressed by astrocytes and microglia [89]. The expression of ITSN1-L was negatively correlated with brain tumor grade and prognosis in glioma. By contrast, expression of ITSN1-S was elevated in glioma [90]. Notably, while these isoforms did not shown differences in promoting cell growth, only ITSN1-L was capable to inhibit cell migration and invasion by affecting HDAC6-mediated stability of microtubules. Moreover, ITSN1-L was shown to attenuate cell-substrate adhesion and strengthen cell-cell junctions [90] (Table 2). These results suggest that a splicing switch from ITSN1-L to ITSN1-S favors glioma progression by modulating pro-invasion features. 

The expression of inflammatory genes modulates the TME in several cancers, including GBM. The tumor suppressor Kruppel-like factor 6 gene (KLF6) encodes for a transcription factor that promotes expression of negative regulators of NF-κB, a master regulator of pro-inflammatory cytokines. Notably, KLF6 is often deleted in GBM [91]. Moreover, KLF6-expressing GBMs often express a dominant-negative splice variant (KLF6-SV1), which is mislocalized to the cytoplasm and antagonizes the tumor suppressor activity of the full-length KLF6 protein (KLF6-FL) [92]. KLF6-SV1 lacks the anti-inflammatory function of the full-length protein and correlates with a more aggressive behavior and drug-resistance in GBM [91,93], suggesting that its impact on cytokine expression favors the establishment of a more malignant TME (Table 2). 

Estrogen-related receptor **β** ERR-**β** (*ESRRB*), a member of orphan nuclear receptor superfamily, is alternatively spliced at the 3’-end, leading to the production of several splicing variants [94]. Among them, ERRβ-2 shows pro-apoptotic features and interacts with the actin nucleation-promoting factor cortactin [95] (Table 2). Cortactin regulates membrane trafficking and the secretion of matrix metalloproteinases (MMPs) that regulate ECM degradation [96]. Moreover, it promotes invasion and migration of glioma cells [97]. Since ERRβ-2 suppresses GBM cell migration, this splice variant may exert its role by limiting the activity of cortactin in GBM cells. Cortactin has been also implicated in release of extracellular vesicles (EVs) in the TME [98]. EVs are key messengers of intercellular communications that control the establishment and maintenance of TME [99], are actively secreted by GBM cells, and promote their oncogenic features [100,101,102,103]. For instance, GBM-secreted EVs stimulate recipient astrocytes to acquire a pro-tumoral phenotype by delivering several factors, including MYC and MYCN [101]. Furthermore, EV-mediated transfer of splicing factors from apoptotic to healthy GBM cells has also been proposed as a mechanism to alter splicing patterns and to promote malignancy. EVs released by apoptotic GBM cells were enriched in core spliceosomal proteins and snRNAs. Once internalized in healthy GBM cells, these factors altered the splicing program of recipient cells and promoted a mesenchymal-like phenotype [102]. In particular, it was shown that the transfer of the splicing factor RBM11 promoted cell proliferation and chemoresistance of surviving GBM cells by altering the splicing of cancer-related genes towards more oncogenic isoforms [102] (Table 2).

Collectively, these observations suggest that splicing dysregulation in brain tumors elicits a direct impact on TME features, which generally favors progression to more aggressive stages.

## 6. Back Splicing in Brain Tumors

Circular RNAs (circRNAs) are endogenous RNAs derived from a back-splicing process (i.e., the covalent joining of a downstream splice donor site with an upstream splice acceptor site). CircRNA biogenesis is mediated by the canonical spliceosome and regulated by the same *cis*-regulatory elements and *trans*-acting factors that control linear splicing [104]. The presence of repetitive elements in an opposite orientation in distant introns, as well as the dimerization of specific RNA binding proteins that bind to intronic sites, have been shown to promote circRNA biogenesis [104]. Since alternative circRNAs can be produced from the same gene, similarly to the splicing of linear transcripts, and circRNA biogenesis utilizes the same machinery and cis-regulatory elements of linear splicing, it can be considered as an additional form of alternative splicing. 

Even if the majority of circRNAs still lack functional annotations, recent observations reveal some potentially important roles in gene regulation [104]. For instance, a set of intron-containing circRNAs regulate gene transcription in *cis* by directly interacting with the elongating RNA polymerase II (RNAPII) complex [105] or with the U1 snRNP [106]. CircRNAs biogenesis can also compete with pre-mRNA splicing, resulting in lower levels of linear mRNAs that include circularized exons [107,108]. Furthermore, circRNAs can function as sponge for microRNAs (miRNAs) [109] or RNA binding proteins, thereby regulating intracellular mRNA fate [107,110,111]. Lastly, some circRNAs, that contain AUG sites and internal ribosome entry site (IRES) elements can be translated in small peptides, thus to expand their regulatory repertoire [112,113,114].

CircRNAs are aberrantly expressed in several types of cancer, where they regulate multiple biological processes [115], including in brain tumors [116,117,118] (Table 2). For instance, circSMARCA5 was identified as one of the circRNAs most differentially expressed in a cohort of fifty-six GBM patient biopsies. CircSMARCA5 expression negatively correlated with histological grade, with a strong down-regulation in higher grade. This observation correlated with the reduced migration capacity/propensity of GBM cells over-expressing circSMARCA5. Interestingly, circSMARCA5 acts as a sponge for the oncogenic splicing factor SRSF1, thus limiting its activity [119,120]. SRSF1 is up-regulated in GBM patients and promotes pro-angiogenic VEGF isoforms [120,121]. Thus, down-regulation of circSMARCA5 in GBM unleashes a *trans*-acting splicing regulator to promote pro-angiogenic features. On the other hand, the circNT5E RNA plays oncogenic functions in GBM. CircNT5E controls multiple pathologic processes, including cell proliferation, migration, and invasion, by directly sponging miR-422a and inhibiting its activity. Furthermore, circNT5E was observed to sponge other miRNAs, suggesting the existence of a more complex regulatory network controlled by circNT5E in GBM [122].

Some circRNAs may also encode proteins, which in turn directly regulate cellular processes in normal brain tissue and glioma. The circFBXW7 RNA encodes a novel 21-kDa protein, called FBXW7-185aa, which reduces the half-life of c-MYC, thus inhibiting the proliferation of GBM cells. Accordingly, expression of circFBXW7 is positively associated with overall survival in GBM patients [123] (Table 2). The circRNA generated by the long intergenic non-protein-coding RNA p53-induced transcript (LINC-PINT) also encodes for a peptide that acts as tumor suppressor by reducing the proliferation rate of GBM cells. Expression of this circRNA is strongly reduced in GBM compared to healthy tissues and the encoded protein directly interacts with the polymerase associated factor complex (PAF1c) and inhibits transcriptional elongation of numerous oncogenes [124] (Table 2). Furthermore, some circRNAs may affect the tumorigenicity and/or the malignant behavior of GBM cells by regulating, directly or indirectly, the activity of their linear counterpart. This is the case of a novel 17-kDa protein, called SHPRH-146aa, encoded by the circRNA produced from the SNF2 histone linker PHD RING helicase gene (*SHPRH*). It is noteworthy that the start AUG codon and the stop UGA codon of the open reading frame used for the SHPRH-146aa peptide overlap, as both use the same A base [125]. Mechanistically, SHPRH-146aa stabilizes the SHPRH full-length protein by inhibiting its proteasome-dependent degradation. In turn, the SHPRH protein reduces proliferation of GBM cells by promoting degradation of the proliferating cell nuclear antigen (PCNA) [125] (Table 2). 

## 7. Therapeutic Approaches

Primary and secondary brain tumors can be classified from grade I to grade IV, according to their proliferative and infiltrative potential (Table 1) [1]. Current therapies include tumor surgical resection, ionizing radiation and chemotherapy. Despite these invasive treatments, the overall outcome is ineffective for most of these tumors. Moreover, predictive and reliable markers of therapy response are urgently needed. 

The association between splicing dysregulation and malignancy suggests that brain tumors might benefit from splicing targeted therapies. To date, several splicing-based pre-clinical approaches are being developed, and some have already entered the clinical practice [16]. In some cases, these approaches have been also applied to brain tumors, eliciting promising pre-clinical results. In particular, ASOs targeting both protein-coding and regulatory non-coding RNA [126,127], sense oligonucleotides acting as decoy for specific splicing factors [128], or small drugs that modulate the activity of the spliceosome [53] have been tested in brain tumor models.

Splice switching oligos (SSOs) are ASOs designed to force the production of selected splicing isoforms. SSOs anneal to the complementary sequence of the target pre-mRNA and inhibit binding of splicing factors or spliceosomal components, thus allowing inclusion/skipping of selected exons. This approach has proved to be an efficient therapeutic strategy in neurodegenerative diseases, such as spinal muscular atrophy [126]. Recently, pre-clinical studies have tested this technology also in brain tumors (Table 2). Alternative selection of the two last exons in the MNK2 kinase gene (*MKNK2*) yields either the MNK2a variant, which displays tumor suppressor activity, or the pro-oncogenic MNK2b variant. In vivo delivery of an SSO that efficiently promotes splicing of MNK2a inhibited GBM growth and sensitized GBM cells to chemotherapeutic agents [129]. Comparable anti-cancer results were obtained by employing the same strategy to promote splicing of a truncated inactive telomerase subunit (hTERT). The hTERT ASO (AON-Ex726) inhibited proliferation and induced apoptosis of GBM cells by reducing the levels of the full-length hTERT isoform [130]. 

ASO that interfere with miRNAs can also indirectly modulate splicing. The expression of mir-10b is up-regulated in both primary and metastatic brain tumors [131]. ASO-mediated reduction of miR-10b levels impaired glioma cell viability without affecting survival of normal brain cells. Interestingly, in this condition the most de-repressed genes were splicing factors that are normally down-regulated in GBM, suggesting that the oncogenic function of miR-10b partially relies on splicing regulation [132] (Table 2). 

Decoy RNA oligonucleotides (DROs) that specifically interfere with the activity of selected splicing factors have been recently developed. DROs are modified RNA molecules composed of a tandem binding motif for a specific splicing factor. The binding of splicing factors to DROs titers them out and inhibits their splicing activity towards endogenous targets [128]. This strategy was successfully used to target the activity of the splicing factors RBFOX1/2, PTBP1, and SRSF1. Importantly, the intracranial injection of an SRSF1-targeted DRO in a GBM mouse model decreased the oncogenic properties of cancer cells by reverting the splicing of oncogenic splice variants of SRSF1-target genes (*INSR*, *U2AF1*, *MKNK2*, *USP8*; see Table 2) [128].

**Table 2 cells-09-00010-t002:** List of the main oncogenic splice variants and splicing-related molecules described in the text.

Spliced Isoform	Tumor Type	Splicing Factor/ASO ^1^/DRO ^2^	Biological Function	Reference
BAF45d/6A-	GBM ^3^, GSC	PTBP1	Cell growth, self-renewal, and tumorigenicity	[44]
PKM2	GBM	PTBP1, hnRNPA1, hnRNPA2	Cell metabolism	[45,46]
ETV1-E7, PUM2-E13, SMN-E6, NDE1-E9, PKM2	GBM, GSC ^4^	SRSF3	Cell growth, self-renewal, metabolism and tumorigenicity	[47,48]
PKM1	GBM	10W_45-59_ and 10M_46-60_ ASOs	Apoptosis	[49]
KAP (d variant)	GBM	unknown	Cell proliferation and migration	[50]
MADD/E16-	GBM	hnRNPH	Cell proliferation and apoptosis	[51]
MYO1B-FL	GBM	SRSF1	Cells proliferation, survival, and invasion	[35]
ΔMAX	GBM	hnRNPA1	Cell metabolism	[62]
ANXA7-I2	GBM	PTBP1	Membrane trafficking	[43]
CD44s	GBM	unknown	EGFR degradation	[66]
TAZ	Glioma	USP39	Cell proliferation	[73]
PMEPA1a	Glioma	unknown	Cell proliferation	[74]
tGLI1	GBM, MB ^5^	unknown	Angiogenesis and cell growth	[79]
L-VEGF_144_a	GBM	unknown	Angiogenesis and therapeutic response	[85,86]
ITSN1-L	Glioma	unknown	Cell migration, invasion and adhesion	[90]
KLF6-SV1	GBM	unknown	Inflammatory response	[91,92,93]
ERRβ-2	GBM	SRSF6	Apoptosis and cell migration	[95]
CyclinD1a, MDM4s, PKM2	GBM	RBM11	Cell proliferation and chemoresistance	[102]
circSMARCA5	GBM	unknown	Cell migration and angiogenesis	[119,120]
circNT5E	GBM	unknown	Cell proliferation, migration, and invasion	[122]
circFBXW7	GBM	unknown	Cell proliferation	[123]
circLINC-PINT	GBM	unknown	Transcriptional elongation	[124]
circSHPRH	GBM	unknown	Cell proliferation	[125]
MNK2a	GBM	2b-block SSO	Cell growth and chemotherapeutic response	[129]
trunc-hTERT	GBM	AON-Ex726 SSO	Cell proliferation and apoptosis	[130]
SRSF1-target genes	GBM	SF2i1 and SF2i2 DROs	p38-MAPK pathway	[128]

^1^ ASO: antisense oligonucleotide; ^2^ DRO: Decoy RNA oligonucleotides; ^3^ GBM: glioblastoma; ^4^ GSC: glioma stem cell; ^5^ MB: medulloblastoma.

These observations suggest that RNA-based therapeutic strategies may be useful to counteract brain tumorigenesis. The selective nature of RNA-RNA interactions makes these molecules highly specific for their targets, thus reducing broad cellular detrimental effects. This is exemplified by the excellent results obtained by Nusinersen, an SSO that rescues exon 7 splicing in the *SMN2* gene, in patients affected by Spinal Muscular Atrophy [126]. However, the major current limit to strategies based on SSOs and DROs is represented by their delivery, as they should be modified to allow for crossing of the blood–brain barrier in order to avoid intrathecal injections in patients. 

Approaches to more globally target splicing regulation by chemical inhibitors for the spliceosome machinery are also being evaluated in brain tumors. In this regard, a promising target is PRMT5, a type II methyltransferase that is responsible for depositing the majority of the symmetrical dimethylation marks on arginines (SDMA) and methylates several RNA binding proteins and spliceosomal proteins [133,134,135]. In MYC-driven tumors, PRMT5 enhances assembly of spliceosome components and MYC expression [136,137]. Since MYC promotes GBM cell survival, proliferation, invasiveness, and self-renewal, its expression is inversely correlated with patient survival [53,138,139,140]. PRMT5 inhibition in GBM causes the accumulation of intron-containing pre-mRNAs, mainly affecting genes involved in cell-cycle progression, chromosome segregation, RNA biology, and DNA repair. These observations suggest that GBM is highly dependent on PRMT5 up-regulation to ensure efficient splicing [53] and that it might be a promising therapeutic target for MYC-driven tumors [141,142] (Figure 3). Indeed, MYC up-regulation induces a pervasive activation of gene transcription and it is likely that cells need to coordinate this transcriptional burden with an efficient RNA processing rate. 

Specific inhibitors of PRMT5 activity have been recently developed. Two of these, EPZ015666 and CMP5, have shown promising results in GBM, as they efficiently reduced tumor growth in both in vitro and in vivo models [53,138]. Moreover, three PRMT5 inhibitors [GSK3326595 (clinicaltrials.gov NCT02783300), PF 06,939,999 (clinicaltrials.gov NCT03854227), and JNJ-64619178 (clinicaltrials.gov NCT03573310)] are in phase I clinical trials for patients with advanced solid tumors [143]. Since non-tumoral cells express low levels of PRMT5, these inhibitors may show limited cellular toxicity and high cancer cell specificity. These features may favor their employment in novel therapeutic approaches for cancers that are unresponsive to conventional therapies, either as single agents or in combined regimens.

## 8. Conclusions

Alternative splicing is a versatile mechanism utilized by eukaryotic cells to finely tune gene expression. As most human genes undergo alternative splicing, virtually every cellular pathway is regulated by this process. Mounting evidence indicates that alternative splicing regulation is a sensitive mechanism employed by cells to integrate the responses to environmental and endogenous cues. Brain tumors, and in particular GBM and SHH-MB, have been recently shown to rely on altered splicing regulation for their onset or progression. At the same time, splicing dysregulation is emerging as a vulnerability of brain cancer cells that can be exploited therapeutically. Herein, we have discussed how deregulated cellular signals drive aberrant pro-oncogenic splicing programs in brain tumors, and the way in which oncogenic splice variants interfere with key cellular processes. Moreover, in some cases, such as SHH-MB, mutations in spliceosome components represent a driver event in tumorigenesis, with broad consequences on the transcriptome of cancer cells. Collectively, the examples described in this study indicate that brain tumors rely on the modulation of alternative splicing to promote and sustain cancer cell proliferation, as well as to shape the tumor environment and favor cancer cell survival. Although a direct comparison of the relative impact of transcription and splicing dysregulation in brain tumors has not been performed yet, it is likely that the two pathways cooperate. Indeed, splicing is mechanistically coupled to transcription and, in turn, splicing factors elicit widespread effects on transcription [144]. This tight connection between the transcription and processing of nascent RNAs needs to be precisely controlled in order to avoid detrimental effects originating from the expression of aberrant gene products. 

The increasing interest in splicing regulation observed in the past decade has clarified many aspects of this sophisticated mechanism. Nevertheless, further efforts are still necessary to clarify the paramount impact of splicing dysregulation in brain tumors and to assess the potential of targeting this process as an efficacious therapeutic approach for these neoplastic diseases. 

## Figures and Tables

**Figure 1 cells-09-00010-f001:**
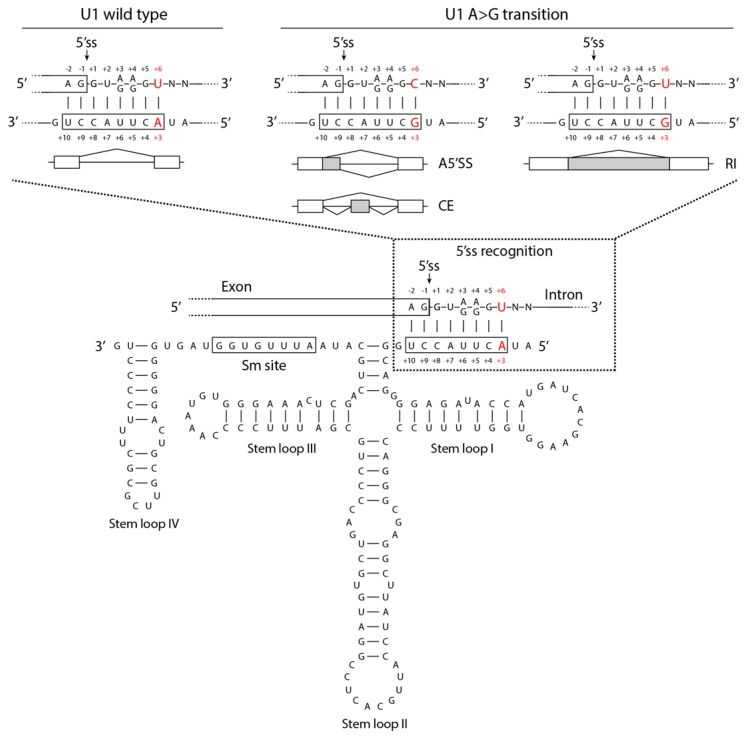
Schematic representation of alternative splicing events mediated by U1 snRNA A>G transition in medulloblastoma. Base-pairing between the wild-type 5′ss recognition sequence (3–10 nt) of U1 snRNA and 5′ss region (+6 to −2) of target pre-mRNA is shown in the dashed box. Wilde-type U1 snRNA ensures proper 5′ss recognition and splicing (left panel). The A>G transition at the third nt of the U1 snRNA favors G-C base-pairing increasing cryptic A5′ss and CE splicing events (middle panel); whereas, G-T base-pair mismatch increases intron retention events (right panel). Secondary structure of the U1 snRNA is represented; the third and the sixth nucleotide of U1 snRNA and intron of pre-mRNA, respectively, are indicated in red. Constitutive (white boxes) and alternative (gray boxes) exons and introns (black lines) are also shown. A5′ss: alternative 5′ss; CE: cassette exon; RI: intron retention.

**Figure 2 cells-09-00010-f002:**
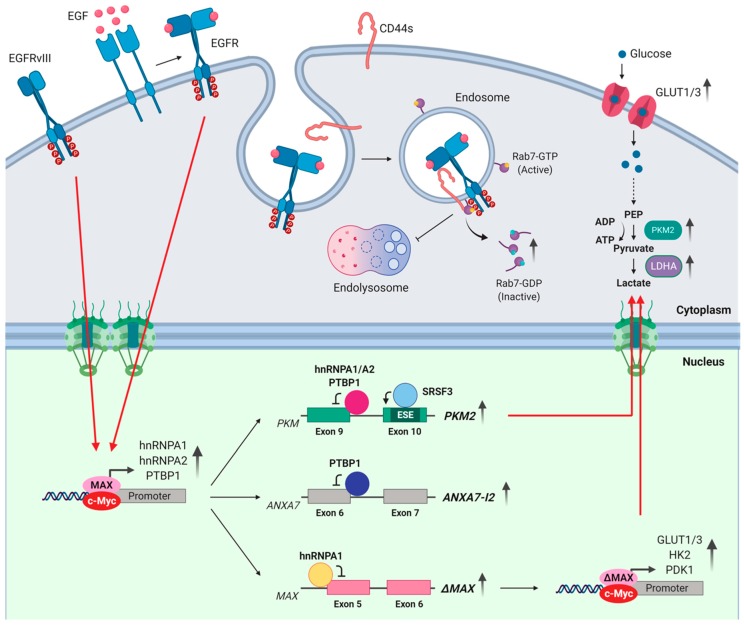
Dysregulation of alternative splicing sustains EGFR/EGFRvIII signaling and metabolism reprogramming in brain tumors. Stimulation of EGFR, and/or expression of the constitutive active EGFRvIII isoform, activates MYC-dependent expression of hnRNP proteins promoting alternative splicing of PKM2 and ΔMAX isoforms. High level of PKM2 and ΔMAX ensures high glycolytic flux and expression of *GLUT1*, *GLUT3*, *PDK1* and *HK2* genes, respectively, sustaining glucose-dependent cell proliferation. Concurrently, CD44s expression and PTBP1-mediated splicing of ANXA7-I2 isoform amplifies EGFR signaling by reducing lysosomal degradation of activated receptor. EGFR: epidermal growth factor receptor; EGFRvIII: epidermal growth factor receptor variant III; EGF: epidermal growth factor; hnRNPA1: heterogeneous nuclear ribonucleoprotein A1; hnRNPA2: heterogeneous nuclear ribonucleoprotein A2; PTBP1: Polypyrimidine tract-binding protein 1; SRSF3: serine and arginine rich splicing factor 3; PKM: pyruvate kinase; PKM2: pyruvate kinase isozymes M2; GLUT1: glucose transporter 1; GLUT3: glucose transporter 3; PEP: phosphoenolpyruvate; ANXA7: annexin A7; ANXA7-I2: annexin A7 isoform 2; PDK1: pyruvate Dehydrogenase Kinase 1; LDHA: lactate dehydrogenase A; HK2: hexokinase 2. MAX: myc-associated factor X; ΔMAX: delta myc-associated factor X.

**Figure 3 cells-09-00010-f003:**
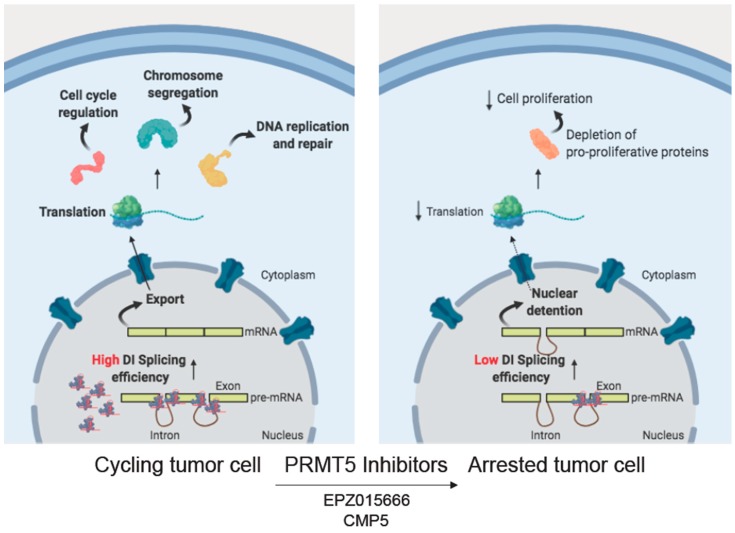
PRMT5 inhibitors as a therapeutic approach for GBM tumors. PRMT5 enhances assembly of spliceosome components in GBM cells, favoring the correct processing of transcripts encoding for proteins mainly involved in survival pathways (cell cycle regulation, chromosome segregation and DNA replication and repair). EPZ015666 and CMP5 dependent PRMT5 inhibition causes the accumulation of intron-containing pre-mRNAs, thus to efficiently reduce tumor growth. DI stands for detained intron.

**Table 1 cells-09-00010-t001:** Grading of CNS tumors mentioned in this review according to WHO guidelines [1].

Tumor Type	Grade
**Diffuse Astrocytic and Oligodendroglial Tumors**	
Diffuse astrocytoma (IDHmut)	II
Oligodendroglioma (IDHmut-1p/19q deleted)	II
Anaplastic astrocytoma (IDHmut)	III
Anaplastic Oligodendroglioma (IDHmut-1p/19q deleted)	III
Diffuse midline Glioma (H3K27Mmut)	IV
Glioblastoma (IDHwt)	IV
Glioblastoma (IDHmut)	IV
**Ependymal Tumor**	
Subependymoma	I
Myxopapillary ependymoma	I
Ependymoma	II
Ependymoma (RELA fusion-positive)	II/III
Anaplastic Ependymoma	III
**Embryonal Tumors**	
Medulloblastoma	IV
Embryonal tumors with multilayered rosette	IV
Medulloepithelioma	IV
CNS Embryonal tumors	IV
Atypical teratoid/rhabdoid tumor	IV
CNS Embryonal tumors with rhabdoid features	IV

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
