# Peer review of "Splicing Dysregulation as Oncogenic Driver and Passenger Factor in Brain Tumors"

_cells, 2019, doi:10.3390/cells9010010_

Round 1
Reviewer 1 Report
This is a nice review of alternative splicing in brain tumors. It is comprehensive and covers all the studies I know.
The review addresses the querstion of whether altered RNA splicing is a feature of brain tumors and whether the changes could be contribute to tumor formation or growth. This is a very active area of research for many tumor types, as it has become apparent that the transcriptomic changes are not simply altered gene expression but involve gene isoform expression changes.
Topic is original. I am not aware of another review on RNA splicing in brain tumors. There are more general reviews on cancer and splicing, but they do not go into depth on different types.
Review is well written and I enjoyed reading it. Conclusions are valid and point to the need for more studies in this area.
Reviewer 2 Report
Bielli et al. describe a systematic review on the emerging literature of brain tumors focusing on splicing dysregulation. The review sounds nice starting from the fundamentals of splicing and well organized except for the two comments below which I believe is necessary to meet the levels to publish in "Cells".
I want to see a simple single table which shows the factors involved in sections 5,6,7 (not for each section but one table for all; it doesn't have to be thorough but a single phrase of the function and reference will help). There are so many factors that it becomes tedious to read when reading these parts. Probably more for the broad audience who reads the review. I want to know if there are any studies that compare the involvement of transcription and splicing in brain tumors.Author Response
Please see the attachment
